# Surface Evaluation of Orthodontic Wires Using Texture and Fractal Dimension Analysis

**DOI:** 10.3390/ma14133688

**Published:** 2021-07-01

**Authors:** Michał Sarul, Marcin Kozakiewicz, Kamil Jurczyszyn

**Affiliations:** 1Department of Integrated Dentistry, Wroclaw Medical University, 50-425 Wroclaw, Poland; 2Department of the Maxillofacial Surgery, Medical University of Lodz, 90-647 Lodz, Poland; marcin.kozakiewicz@umed.lodz.pl; 3Department of Dental Surgery, Wroclaw Medical University, 50-425 Wroclaw, Poland; kamil.jurczyszyn@umed.wroc.pl

**Keywords:** orthodontic wire, orthodontic wire surface, fractal dimension analysis, texture analysis

## Abstract

Mechanical properties of orthodontic wires can have a very significant impact both on the resistance of the entire appliance to the oral cavity conditions and directly on the effectiveness of the therapy. Striving to achieve repeatability of mechanical characteristics of orthodontic wires of a given type should be an obligatory condition in their production. To achieve it, these components should be thoroughly analyzed using various mechanical tests. Twenty-four steel and nickel-titanium orthodontic wires from four different manufacturers were examined. Each wire was subjected to fractal dimension analysis and texture analysis. The two sides of each wire were compared against each other, as well as in terms of variation in the surface area for each wire type made by different manufacturers. Most wires showed significant variation in fractal dimension and texture, both when comparing two sides of the same wire and between individual wires of a given type made by a single manufacturer. When conducting research and clinically using orthodontic wires made of Ni-Ti alloys and stainless steel, it should be assumed that the surface of orthodontic wires shows a significant degree of variation, and wires of the same type from the same manufacturer may differ significantly in this respect.

## 1. Introduction

Since the early 20th century, fixed edgewise appliances have been one of the tools in orthodontic therapy. The main components of such appliances include orthodontic brackets attached directly to the teeth and orthodontic wires connecting them. The outcome of such therapy is highly dependent on the mechanical interaction in the orthodontic archwire-bracket space. Thus, a steady improvement in the mechanical properties of these components has been observed for more than a century [1].

In orthodontic treatment with fixed appliances, wires made of three different materials are widely used: chrome-nickel stainless steel, β-titanium, and nickel-titanium alloy. Components made of β-titanium are usually used for a short time in the final stages of treatment. On the other hand, nickel-titanium alloy (NiTi) and stainless steel (SS) archwires are used successively in the initial stage of treatment as aligning wires or, the latter, in the main stage of treatment as main working ones [2]. Consequently, these components stay in the mouth for a long time. They are also the main active elements for most of the treatment time. Their mechanical properties can therefore have a very significant impact both on the resistance of the entire appliance to the oral cavity conditions and directly on the effectiveness of the therapy. For this reason, striving to achieve repeatability of mechanical characteristics of orthodontic wires of a given type should be an obligatory condition in their production. To achieve it, these components should be thoroughly analyzed using various mechanical tests [2].

Strength, elasticity, and surface feature testing are mentioned as the main mechanical property tests for orthodontic wires. The latter are most often analyzed using tools such as SEM micrography, laser spectroscopy, profilometry, X-ray diffraction analysis, or atomic force microscopy. However, each of these tests requires complex and expensive apparatus [3]. In addition, the geometric parameters commonly used and obtained from them, such as the height of the roughness or the average deviation from the mean line of the profile, do not reflect the complexity of the geometric structure of the surface, nor do they always allow to infer its performance characteristics [4,5].

As a relatively simple and conveying information about the complexity of the geometric structure of the surface, the technique of computer image analysis and the fractal dimension value can be used. Fractal and multifractal surface characteristics have been determined for many materials, among which we should mention: metallic materials and their alloys, ceramic, polymeric, and amorphous materials [6,7,8,9].

One of the methods used in the analysis of digital images is also texture analysis. The texture represents the image properties such as directional (pattern direction) or porosity. The texture is an inhomogeneous property area in an image. On this basis, it is possible to distinguish two images from each other and define areas in a given image that meet certain conditions. The texture represents the regular features of the object’s surface—looking at the image, we can tell whether it represents a smooth object (e.g., a glass surface) or a rough one (e.g., tree bark), as well as whether the presented pattern is more or less regular. One decides the type of texture based on the observation of certain small patterns whose regular arrangement allows for classification [10,11].

This study aims to evaluate the repeatability of orthodontic wire surface properties using fractal analysis. 

The null hypothesis was that the test wires would show no differences in fractal dimension and texture within each tested wire between the two opposing sides and between the tested wires within the group.

## 2. Materials and Methods

Nickel-titanium and steel orthodontic wires with a cross-section of 0.019 × 0.025 inches from four different manufacturers: Adenta GmbH (Gilching, Germany), Forestdent, GmbH (Pforzheim, Germany), Ormco (Brea, CA, USA), G&H Orthodontics (Cleveland, IN, USA), labeled sequentially, were selected for the study:
NiTi wiresAdenta—ANForestadent—FNOrmco—ONG&H—GNChrome-nickel stainless steel wires:Adenta—ASForestadent—FSOrmco—OSG&H—GS


For each wire type, one wire was randomly selected from three randomly selected packages. Each wire was tested at 15 points on the wider surface of the wire, marked A1, A2, A3, B1, B2, B3, C1, C2, C3, D1, D2, D3, E1, E2, E3: A1‘, A2‘, A3‘, B1‘, B2‘, B3‘, C1‘, C2‘, C3‘, D1‘, D2‘, D3‘, E1‘, E2‘, E3‘ (Figure 1). Prim points are located on the second side of the wire.

### 2.1. Taking Pictures

All photos were taken using a stereoscopic microscope Techrebal K10E (Techrebal, Wilczyce, Poland). The eyepiece was replaced by a ZWO ASI178 mm monochrome digital camera (ZWO CO., LTD, Suzhou, China). All photos were taken applying 36× magnification. Depending on the wire material, the exposure time was set to achieve histogram filling at the 90% range. The gain parameter (sensitivity of CMOS matrix) was the same during all procedures and set for 10 to reduce noise. We used the 14-bit mode of the camera to achieve the widest dynamic range of photos. Images were saved as 16-bit TIFF (Tagged Image File Format) files. While the photos were taken, the wires were on a black and white mosaic. This pattern allows setting white and black points to equalize histograms. All images were cropped to 2800 × 1000 pixels and saved as 8-bit grayscale bitmaps. All graphic operations were performed using GIMP version 2.10.24 (GNU Image Manipulation Program—www.gimp.org (accessed on 3 May 2021), free and open-source license).

### 2.2. Fractal Dimension Analysis

All fractal analyses were performed in ImageJ version 1.53e (Image Processing and Analysis in Java—Wayne Rasband and contributors, National Institutes of Health, Bethesda, MD, USA, public domain license, https://imagej.nih.gov/ij/ accessed on 3 May 2021) and plugin FracLac version 2.5 (Charles Sturt University, Bathurst, Australia, public domain license). In our study, we calculated the FD with the following options of FracLac: grayscale images, differential mode, black locked for background, scaling method: power series.

In the classic counting box method of fractal dimension analysis, source images must be a one-bit bitmap (1 for pixel on and 0 for pixel off). Fractal dimension (D_S_) is calculated using the following formula:(1)$$Ds=\underset{\varepsilon \to 0}{\lim}\frac{\log N(\varepsilon )}{\log\left(\frac{1}{\varepsilon }\right)}$$
where Ds—fractal dimension (FD); ε—length of the box that creates a mesh covering the surface with the examined pattern; N(ε)—minimal number of boxes required to cover the examined pattern.

Conversion of 8-bit images into 1-bit leads to a decrease of details. In our study, we decided to use a modified algorithm of the box-counting method, which allows analyzing monochromatic images such as 8- or 16-bit ones. In the case of grayscale images, FracLac offers three options of FD analysis. One of these options is intensity difference which we applied in our study. The analyzed image is divided into squares (Figure 2a). The difference between maximum pixel intensity and minimum pixel intensity is counted in each square (δIi,j,ε, where i, j—location of the analyzed square in a scale ε):

δIi,j,ε = maximum pixel intensity i,j,ε— minimum pixel intensity i,j,ε


In the next step, 1 is added to the intensity difference to prevent its value being 0:

Ii,j,ε = δIi,j,ε + 1


Finally, the fractal dimension of the intensity difference is described by the following formula (Figure 2c):(2)$$D\;Idiff=\underset{\varepsilon \to 0}{\lim}\frac{\ln\left(I\varepsilon \right)}{\ln\left(\frac{1}{\varepsilon }\right)}$$
where D Idiff is the intensity difference fractal dimension, Iε = Σ(1δIi,j,ε + 1), and ε is the scale of the square. All operations are shown in Figure 2.

The surface texture of orthodontic wires was evaluated using features derived from two groups (run-length matrix and co-occurrence matrix) and previously described texture index (TI) [12]. The regions of interest (ROIs) were normalized (μ ± 3σ) to share the same average (μ) and standard deviation (σ) of optical density within the ROIs. Selected image texture features (entropy and difference entropy from the co-occurrence matrix and long-run emphasis moment from the run-length matrix) in ROIs were calculated for reference bone and bone with collagen scaffold applied:(3)$$Entropy=-\overset{Ng}{\underset{i=1}{\sum }}\overset{Ng}{\underset{j=1}{\sum }}p\left(i,j\right)log(p\left(i,j\right)$$
(4)$$DifEntr=-\sum_{i=1}^{Ng}p_{x-y}\left(i\right)log\left(p_{x-y}\left(i\right)\right)$$
where Σ is the sum, Ng is the number of optical density levels in the radiograph, i and j are the optical density of pixels that are 5 pixels away from one another, p is probability, and log is the common logarithm [13],
(5)$$LngREmph=\frac{\sum_{i=1}^{Ng}\sum_{k=1}^{Nr}k^{2}p\left(i,k\right)}{\sum_{i=1}^{Ng}\sum_{k=1}^{Nr}p\left(i,k\right)}$$
where Σ is the sum, Nr is the number of series of pixels with density level i and length k, Ng is the number of levels for image optical density, Nr is the number of pixels in series, and p is probability [14,15]. Long run-length emphasis moment (LngREmph) was computed from data taken along the long axis of the wire, and measures of disarrangement (entropy and difference entropy, i.e., DifEntr) were computed as non-directional measures. Two of three equations were subsequently used for the texture index construction. Finally, the texture index (TI), which represents the ratio of the measure of the diversity of the structure observed in the radiograph to the measure of the presence of uniform longitudinal structures, was calculated:(6)$$Texture\;Index=\frac{Entropy}{LngREmph}=\frac{(-\sum_{i=1}^{Ng}\sum_{j=1}^{Ng}p\left(i,j\right)log\left(p\left(i,j\right)\right))\sum_{i=1}^{Ng}\sum_{k=1}^{Nr}p\left(i,k\right)}{\sum_{i=1}^{Ng}\sum_{k=1}^{Nr}k^{2}p\left(i,k\right)}$$

### 2.3. Statistical Analysis

Statistica version 13.3 (StatSoft, Cracow, Poland) was used to perform all statistical tests. A statistical significance level of 0.05 was assumed. The Shapiro–Wilk test was used to confirm the normality of distribution. Due to the normal distribution of samples, we performed parametric tests. Student’s *t*-test was performed to check statistical differences between various materials of the same manufacturer. Analysis of variance (ANOVA) and post hoc least significant difference were applied to reveal fractal dimension differences between surfaces of wires of the same type and between producers. The correlation matrix was used to estimate the correlation of FD between two surfaces of the same wire.

Texture comparisons between wire sides and material were performed with one-way ANOVA or the Kruskal–Wallis test depending on the presence of normal distribution. Simple regression analysis was also performed to investigate relationships between general mineral condition parameters and radiological texture features. When *p* < 0.05, the difference was considered statistically significant. Stargraphics Centurion 18 ver.18.1.12 (StarPoint Technologies, Inc., Addison, TX, USA) was used for statistical analyses.

## 3. Results

The graph (Figure 3) shows a comparison of the averaged fractal dimension data for the two sides of each wire. A comparison of the statistical significance of the differences in the fractal dimensions of the individual wires concerning the side measured is shown in Table 1.

Table 2 presents Pearson’s correlation coefficients for the fractional dimension values of individual wire surfaces. We observed positive linear correlation in FS and FN wires.

Comparisons of the degree of variation in surface roughness, as determined by fractal dimension, averaged over the entire wire group (understood as the manufacturer’s brand) in the nickel-titanium and steel wire groups are shown in Table 3.

Graphical results of the surface texture evaluation are shown in Figure 4. Note that the texture feature LngREmph has preserved the directionality property, i.e., the analysis was performed horizontally (along the wire). For the co-occurrence feature matrix, the directionality was reduced (arithmetic mean of the four directions) to a directionless representation of the surface texture appearance.

A plot representation of the surface structure of stainless-steel wires is shown in Figure 5. NiTi archwires have less developed surface structure (*p* < 0.05) in comparison with steel wires (DifEntr, entropy, texture index): TI for NiTi is 0.50 ± 0.17 and for steel is 0.28 ± 0.17. They are also noticeably (*p* < 0.05) less longitudinally scratched (LngREmph) (Table 3). The most homogeneously distributed surface texture features (high texture index values) were observed on Adenta NiTi wires (Figure 6).

NiTi archwires have a less developed surface structure (*p* < 0.05) in comparison with steel wires (DifEntr, entropy, texture index): TI for NiTi is 0.50 ± 0.17 and for steel it is 0.28 ± 0.17. They are also noticeably (*p* < 0.05) less longitudinally scratched (LngREmph) (Table 3). The most homogeneously distributed surface texture features (high texture index values) were observed for Adenta NiTi wires (Figure 5 and Figure 6).

## 4. Discussion

After averaging the values of the measurements taken for the entire surface of one side of a given wire and comparing it with the measurement for the opposite side of the given wire, no statistically significant differences were found only for AN and FS wires. For wires from other manufacturers, at least one of the wires tested showed a statistically significant difference between the two opposite sides of the wire in terms of mean fractal dimension. The FS wires exhibited the most uniform surface, with no differences in fractal dimension, for any of the measured surfaces, within each wire or between the surfaces of individual wires. For all of the wires measured, except the FS wires, there were statistically significant differences in fractal dimension between the mean values of one wire from a given manufacturer and any of the surfaces of another wire within the group. Furthermore, significant and worth noting is the fact that statistically significant differences were more common in steel wires (except for the FS wires mentioned above), i.e., in terms of fractal dimension, surfaces tended to vary more within a single wire and between wires from one manufacturer. The OS wires showed the most non-uniform surface of all the wires—only two measurements out of 15 taken for all surfaces showed no statistically significant differences. Furthermore, the correlation measurement showed that all three tested correlations of the fractal dimension values of individual FS wire surfaces and the averaged correlation dimension were in the >0.5 range. The situation was similar for two of the three studied and the mean of all measurements for the correlation of the fractal dimension values of the FN wire surfaces. For the other wires, the correlation coefficient ranged from −0.5 to 0.5, and for the surface measurement of sample No. 2 of AN wires, it was less than −0.5. In summary, it can be concluded that only the Forestadent wires showed a high uniformity in the surface fractal dimension values of both steel and NiTi wires. The remaining wires showed a large or very large variation in the fractal dimension of the surfaces of the measured wires and thus a large variation in the types of surface roughness.

An important consideration is to determine whether the observed changes may be of clinical and research significance.

The surface topography of orthodontic wires can affect friction released in the orthodontic wire/orthodontic bracket system, the degree of bacterial adhesion, corrosion resistance, and the degree of ion release, especially nickel ions [16,17,18].

The friction triggered in the orthodontic wire/orthodontic bracket system is of particular importance during the aligning stage with nickel-titanium archwires and during the space closure stage with sliding mechanics, which in turn uses steel wires. Orthodontic treatment involves the use of minimal, effective orthodontic forces. Nevertheless, in the straight-wire technique, the forces released by the orthodontic wire are modified by the value of the frictional force released between the wire and the orthodontic bracket [19]. The surface topography of orthodontic wires is cited as one of the main factors affecting the value of the resulting frictional force in the orthodontic wire/orthodontic bracket system, both in the aligning and main stages. Thus, the lack of homogeneity in the surface topography of the wires may affect the actual biomechanics of tooth displacement during fixed appliance treatment [19,20]. Among the wires tested, most—excluding FS and NS wires—showed significant differences in surface topography. These differences involved not only comparisons between different wires from the same manufacturer but even opposite surfaces of the same wire. Thus, the presented studies describe one of the phenomena which make it impossible to assume that the friction values generated in the orthodontic wire/orthodontic bracket system and determined in laboratory tests are standard values for the given type of wire.

The lack of homogeneity of the surface topography of the tested orthodontic wires, confirmed in the study, may also be related to the degree of adhesion of the bacterial plaque [21,22]. The authors showed a lack of homogeneity in the surface structure even between the individual sides of the same wire. Moreover, in the presented study, using the analysis of textures and fractal dimensions, it was found that the factor of the wire producer has a greater impact on the homogeneity of the surface structure than the material from which the wire is made. It should be considered whether the above assumptions should not be taken into account when studying the degree of adhesion of bacterial biofilm to the surface of orthodontic wires. In addition, it is further evidence to what extent the perfection of the orthodontic component manufacturing process can be clinically relevant.

Corrosion and fracture susceptibility and ion release are further factors, which are interdependent and largely due to the surface topography of orthodontic wires. According to available studies, the release rate of these ions and corrosion level can be much higher for wires with a rough surface [23,24,25]. In this context, manufacturers should strive to achieve as smooth a surface as possible, uniform for all surfaces of all wires produced. In this study, the only such wires were from the FS and NS groups. A large variation in surface area was observed for the other wires. In this context, it should be considered whether the lack of uniformity in the surface structure resulting from imperfections in the production processes may affect the degree of corrosion and the release of ions from these wires.

Most studies report that steel wires present a smoother surface to nickel-titanium ones [26,27]. In the fractal dimension study presented here, no such relationship was observed. The only exception was the FS wires, which are steel wires that actually showed a much more uniform and generally smoother surface. The other steel wires showed a significant degree of surface topography variation, and the OS wires had the most surface variation of all the wires tested. It should be noted that the test used by the authors does not explicitly inform the depth of the inequalities present, but rather their type; moreover, it allows a numerical comparison of the degree of uniformity of these inequalities between surfaces. However, the statistical comparison showed a significant degree of surface variation and a complete lack of uniformity in the surface irregularities present on all steel wires, except FS wires, to a greater degree than in the nickel-titanium wire group.

Texture analysis revealed that the quality of workmanship (repeatability of the texture of the two wire surfaces) leaves much to be desired in the steel products of Adental and Ormco and the nickel-titanium products of Forestadent. In this respect, G&H wires show good quality.

Considering that lighter areas in microphotographs of wire surfaces represent smoother areas, it should be emphasized that products with higher LngREmph values have more scratches with smooth peaks. The remaining three variables describing the degree of chaotic fragmentation of surface patterns can be understood as describing its micro-smoothness: the higher DifEtrp, entropy over TI, the smoother the wire surface.

In the light of the texture analysis, it should be pointed out that orthodontic arches made of NiTi are smoother than steel arches. Texture analysis revealed many longitudinal scratches on the surfaces of steel arches. There is a high probability that such a surface image may increase friction during tooth movement and aggravate corrosion in the oral cavity. This is consistent with other observations [28,29]. Therefore, from this point of view, the surface of AN wires should exacerbate these phenomena to the least extent. 

Texture and fractal dimension analysis has not been widely used to determine the surface structure of orthodontic wires. Most analyses use other techniques, such as profilometry, SEM analysis, or atomic-force microscopy. Nevertheless, our analysis allows us to draw similar conclusions, confirmed mathematically, as the research of other authors. Numerous studies to date have shown that orthodontic wires show a significant degree of differentiation in the surface structure, if we take into account wires of a given type supplied by one producer. This applies to both nickel-titanium and steel wires [18,19,26,30,31,32]. Regarding the latter, the studies conducted so far have identified mainly longitudinal scratches on the material surface [31]. In the presented study, the authors obtained very similar results in all of the above-mentioned aspects. In addition, it has been shown that the lack of unification of the structure occurs even within each individual wire, and not only between the wires. The only result that distinguishes the results obtained by the authors is the demonstration that it is possible to obtain and produce wires which have mathematically proven very high uniformity of the surface structure, and thus the occurrence of surface unevenness, but with a very unified structure. This proves that the appropriate care of the manufacturer to maintain repeatable conditions can be an important step on the way to obtaining wires with the most predictable properties in terms of surface structure.

## 5. Conclusions

Most of the wires tested showed a high degree of variation in surface topography;Most of the wires tested showed completely different surface profiles both when comparing individual wires from the same manufacturer and different surfaces of the same wire;In the light of fractal dimension and texture analysis, nickel-titanium wires did not show significantly more variation in surface topography compared to steel wires;When conducting research and clinically using orthodontic wires made of Ni-Ti alloys and stainless steel, it should be assumed that the surface of orthodontic wires shows a significant degree of variation, and wires of the same type from the same manufacturer may differ significantly in this respect.

## Figures and Tables

**Figure 1 materials-14-03688-f001:**
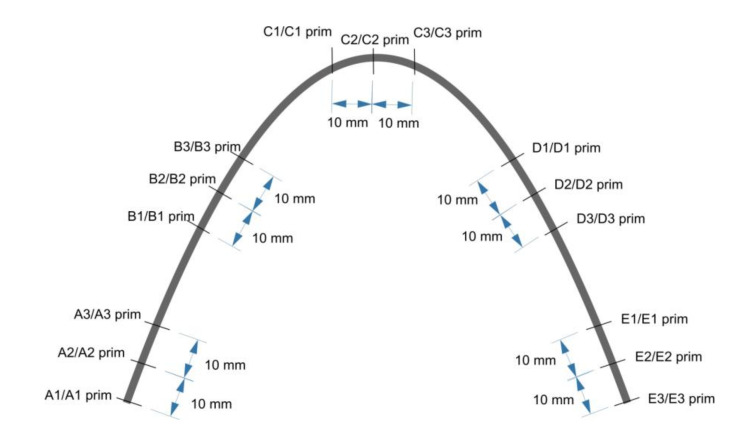
Locations of measurement points (prim—same location on the second side of wire).

**Figure 2 materials-14-03688-f002:**
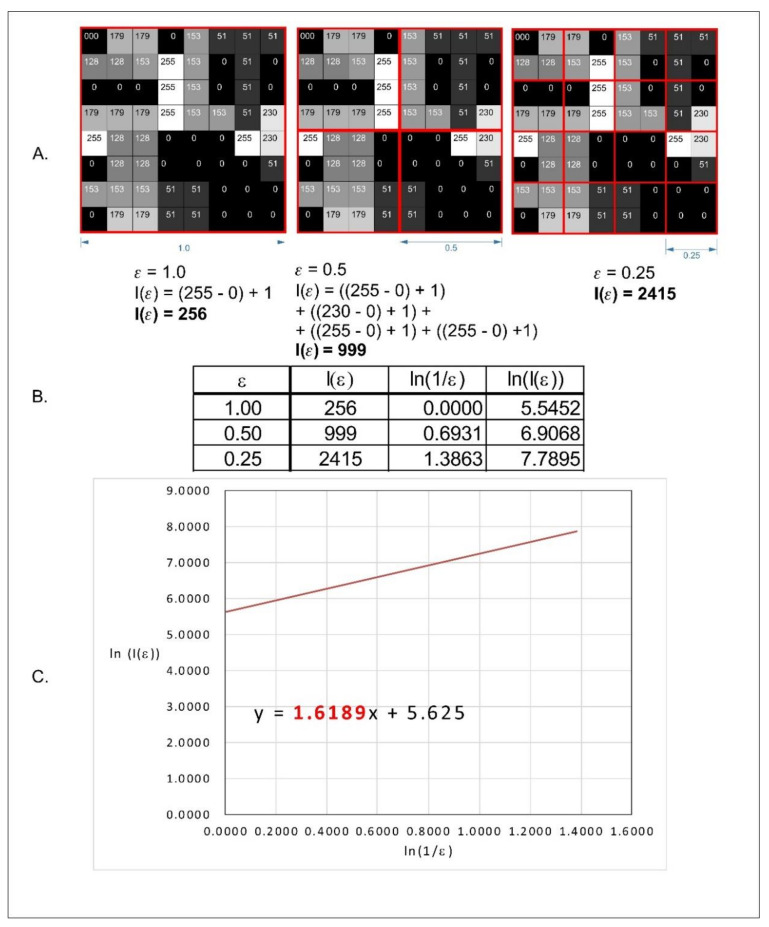
Graphical interpretation of intensity difference algorithm of fractal dimension counting. (**A**) Grayscale 8 bits analyzed image, numbers in squares represent intensity level of each pixel, 0 is black, 255 is white. Red squares represent scale—ε. (**B**) Values of intensity difference for each step of scale reduction (ε = 1.0, ε = 0.5, ε = 0.25). (**C**) Straight line drawn through points from table B on the x–y chart in natural logarithm scale. Slope factor of this straight line is a value fractal dimension calculated by the intense difference algorithm.

**Figure 3 materials-14-03688-f003:**
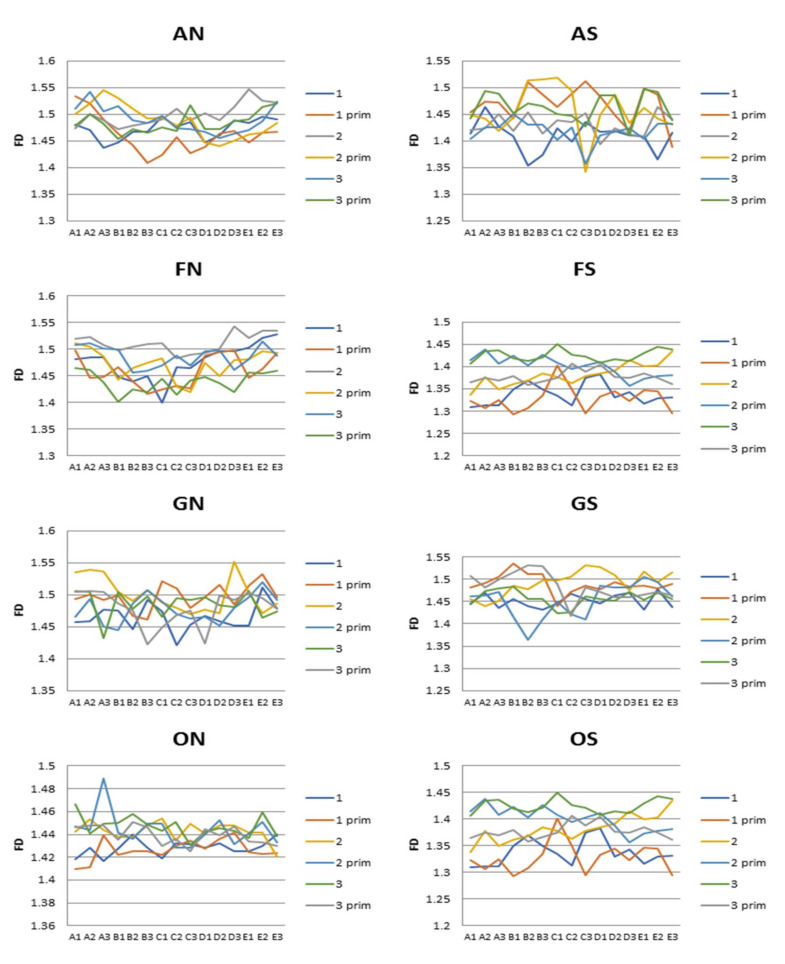
Values of fractal dimension on all measurement points through the length of wires (FD—fractal dimension, A1–E3—measure points, AN—Adenta NiTi, FN—Forestadent NiTi, ON—Ormco NiTi, GN—G&H NiTi, AS—Adenta Steel, FS—Forestadent Steel, OS—Ormco Steel, GS—G&H Steel).

**Figure 4 materials-14-03688-f004:**
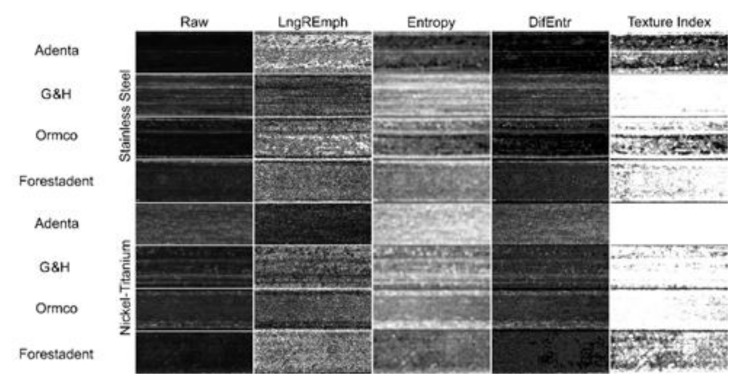
Texture analysis. Surface image transformations of example orthodontic wire samples made of stainless steel and nickel-titanium. Raw (microphotography): Light microscope images. The brighter places correspond to flat smooth areas that reflect visible light more strongly due to this surface. LngREmph: A feature of the series length matrix that reveals areas that similarly reflect incident light. White areas (high values of this texture feature) correspond to elongated horizontal micro-reflections of similar light. This feature does not differentiate continuous dark areas from continuous light areas. Entropy: Reveals a surface with evenly scattered areas of similar brightness (smoothness). Note that Entropy remains in opposition to LngREmph, i.e., areas of high LngREmph feature values occur in areas of low Entropy value, and vice versa. DifEntr: Enhances (shows as whiter) areas made up of evenly scattered areas similarly reflecting incident light (similar surface smoothness). This feature more strongly differentiates the analyzed surface into poorly differentiated areas (black) and highly differentiated areas with a chaotic arrangement of smooth plots (white). By comparing the appearance of this map with the photographic image it can be concluded that the smooth areas of the photographic image correspond to an accumulation of areas of high Difference Entropy. Texture Index (TI): the white areas represent high TI values on the wire surface in the map. The darker the area, the lower the TI value represented by that region. The texture of the wire surface is represented by a lighter area the smoother areas originally entered the archwire photographic image at that location (see next figures), magnification 36×.

**Figure 5 materials-14-03688-f005:**
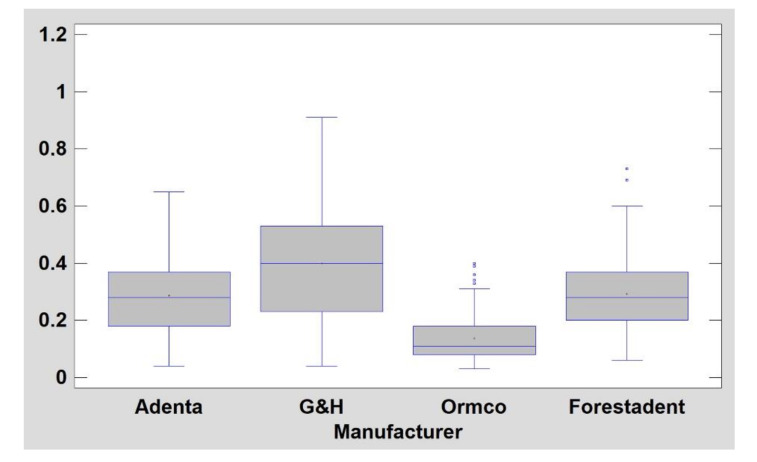
Stainless steel archwire comparison (mean side 1 and side 2 texture index, TI; higher TI means better result). There are significant differences between product groups in the arrangement of texture made up of bright areas (i.e., low-friction surfaces). The best surface is represented by G&H (*p* < 0.05). Slightly worse surfaces for Adenta and Forestadent, which in turn have a significantly higher TI than the Ormco product (*p* < 0.05), y axis indicates values of texture index.

**Figure 6 materials-14-03688-f006:**
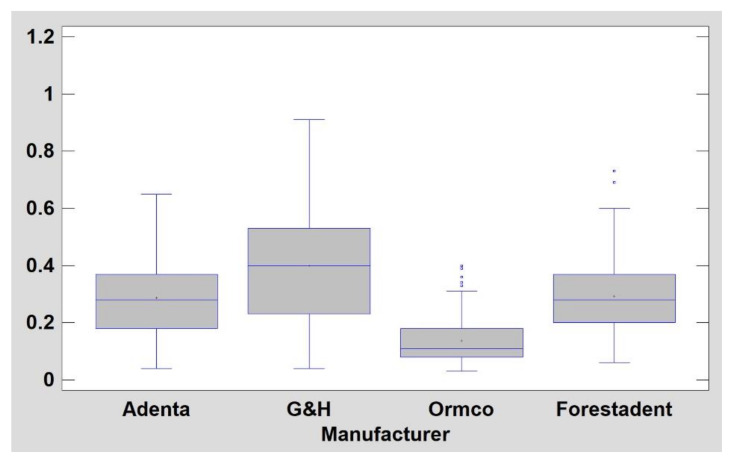
NiTi archwire comparison (mean side 1 and side 2 texture index, TI; higher TI means better result). Adenta wires are significantly (*p* < 0.05) smoother than the last 3 manufacturers’ products. G&H are significantly better (*p* < 0.05) than Ormco and Forestadent wires. Moreover, Forestadent wire is worse than all other products (*p* < 0.05), y axis indicates values of texture index.

**Table 1 materials-14-03688-t001:** Results of least significant difference ANOVA post hoc.

AN						
	1	2	3	1 prim	2 prim	3 prim
1		**0.005910**	0.062750	0.219148	0.112036	0.196845
2	**0.005910**		0.350568	**0.000109**	0.226348	0.131314
3	0.062750	0.350568		**0.002448**	0.780133	0.560056
1 prim	0.219148	**0.000109**	**0.002448**		**0.005594**	**0.012962**
2 prim	0.112036	0.226348	0.780133	**0.005594**		0.761085
3 prim	0.196845	0.131314	0.560056	**0.012962**	0.761085	
AS						
	1	2	3	1 prim	2 prim	3 prim
1		0.064518	0.420929	**0.000001**	**0.000063**	**0.000006**
2	0.064518		0.290185	**0.000824**	**0.021607**	**0.003856**
3	0.420929	0.290185		**0.000019**	**0.001016**	**0.000119**
1 prim	**0.000001**	**0.000824**	**0.000019**		0.262137	0.620087
2 prim	**0.000063**	**0.021607**	**0.001016**	0.262137		0.529513
3 prim	**0.000006**	**0.003856**	**0.000119**	0.620087	0.529513	
FN						
	1	2	3	1 prim	2 prim	3 prim
1		**0.000253**	0.266811	0.054283	0.665814	**0.000085**
2	**0.000253**		**0.008287**	**0.000000**	**0.000054**	**0.000000**
3	0.266811	**0.008287**		**0.002885**	0.124590	**0.000001**
1 prim	0.054283	**0.000000**	**0.002885**		0.132653	**0.032202**
2 prim	0.665814	**0.000054**	0.124590	0.132653		**0.000389**
3 prim	**0.000085**	**0.000000**	**0.000001**	**0.032202**	**0.000389**	
FS						
	1	2	3	1 prim	2 prim	3 prim
1	n.s.	n.s.	n.s.	n.s.	n.s.	n.s.
2	n.s.	n.s.	n.s.	n.s.	n.s.	n.s.
3	n.s.	n.s.	n.s.	n.s.	n.s.	n.s.
1 prim	n.s.	n.s.	n.s.	n.s.	n.s.	n.s.
2 prim	n.s.	n.s.	n.s.	n.s.	n.s.	n.s.
3 prim	n.s.	n.s.	n.s.	n.s.	n.s.	n.s.
GN						
	1	2	3	1 prim	2 prim	3 prim
1		**0.000066**	**0.019560**	**0.000252**	0.124506	0.109532
2	**0.000066**		0.072394	0.707309	**0.009663**	**0.011544**
3	**0.019560**	0.072394		0.152802	0.409625	0.447660
1 prim	**0.000252**	0.707309	0.152802		**0.025675**	**0.030138**
2 prim	0.124506	**0.009663**	0.409625	**0.025675**		0.947698
3 prim	0.109532	**0.011544**	0.447660	**0.030138**	0.947698	
GS						
	1	2	3	1 prim	2 prim	3 prim
1		**0.000066**	0.520221	**0.000181**	0.966604	**0.001564**
2	**0.000066**		**0.000621**	0.777399	**0.000076**	0.353605
3	0.520221	**0.000621**		**0.001547**	0.547652	**0.010335**
1 prim	**0.000181**	0.777399	**0.001547**		**0.000210**	0.518005
2 prim	0.966604	**0.000076**	0.547652	**0.000210**		**0.001783**
3 prim	**0.001564**	0.353605	**0.010335**	0.518005	**0.001783**	
ON						
	1	2	3	1 prim	2 prim	3 prim
1		**0.000105**	**0.000001**	0.606802	**0.000022**	**0.002075**
2	**0.000105**		0.241662	**0.000016**	0.674736	0.374517
3	**0.000001**	0.241662		**0.000000**	0.450552	**0.041336**
1 prim	0.606802	**0.000016**	**0.000000**		**0.000003**	**0.000391**
2 prim	**0.000022**	0.674736	0.450552	**0.000003**		0.192448
3 prim	**0.002075**	0.374517	**0.041336**	**0.000391**	0.192448	
OS						
	1	2	3	1 prim	2 prim	3 prim
1		**0.000000**	**0.000000**	0.260709	**0.000000**	**0.000003**
2	**0.000000**		**0.000001**	**0.000000**	**0.020595**	0.590728
3	**0.000000**	**0.000001**		**0.000000**	**0.002760**	**0.000000**
1 prim	0.260709	**0.000000**	**0.000000**		**0.000000**	**0.000000**
2 prim	**0.000000**	**0.020595**	**0.002760**	**0.000000**		**0.004764**
3 prim	**0.000003**	0.590728	**0.000000**	**0.000000**	**0.004764**	

(AN—Adenta NiTi, FN—Forestadent NiTi, ON—Ormco NiTi, GN—G&H NiTi, AS—Adenta Steel, FS—Forestadent Steel, OS—Ormco Steel, GS—G&H Steel, n.a.—not available in case of *p* > 0.05 in ANOVA test. Bold and underlined font—significant difference *p* < 0.05).

**Table 2 materials-14-03688-t002:** Pearson’s correlation coefficient value between each side of the wire.

Pearson’s Correlation Coefficient Value				
wire type	1 vs. 1 prim	2 vs. 2 prim	3 vs. 3 prim	mean
AN	−0.107	**−0.527**	0.118	−0.172
AS	−0.244	−0.172	−0.241	−0.219
FN	**0.603**	**0.762**	0.440	**0.601**
FS	**0.581**	**0.618**	**0.838**	**0.679**
GN	0.158	−0.063	−0.125	−0.010
GS	−0.191	0.012	0.322	0.048
ON	0.051	0.253	0.451	0.252
OS	−0.256	−0.634	−0.135	−0.341

Bold and underlined font—average and strong linear correlation (FD—fractal dimension, A1–E3—measure points, AN—Adenta NiTi, FN—Forestadent NiTi, ON—Ormco NiTi, GN—G&H NiTi, AS—Adenta Steel, FS—Forestadent Steel, OS—Ormco Steel, GS—G&H Steel).

**Table 3 materials-14-03688-t003:** Comparison of the surface texture of archwires.

Manufacturer	Material	LngREmph		Difference Entropy		Entropy		Texture Index	
		Side 1	Side 2	Side 1	Side 2	Side 1	Side 2	Side 1	Side 2
Adenta	NiTi	4.65 ± 0.96 *	5.29 ± 1.62 *	1.06 ± 0.03 *	1.05 ± 0.05 *	2.79 ± 0.07 *	2.78 ± 0.10 *	0.63 ± 0.16 *	0.58 ± 0.20 *
	Steel	9.01 ± 2.45 *	15.89 ± 13.37 *	0.88 ± 0.04 *#	0.97 ± 0.08 *#	2.44 ± 0.08 *#	2.49 ± 0.09 *#	0.29 ± 0.07 *	0.28 ± 0.19 *
Forestadent	NiTi	5.80 ± 2.01 *#	8.51 ± 4.07 *#	0.93 ± 0.08 #	0.89 ± 0.08 *#	2.51 ± 0.13 *	2.44 ± 0.19 *	0.49 ± 0.18 *#	0.35 ± 0.15 #
	Steel	11.36 ± 6.34 *	10.22 ± 3.74 *	0.99 ± 0.06	0.97 ± 0.05 *	2.65 ± 0.13 *	2.63 ± 0.11 *	0.30 ± 0.15 *	0.29 ± 0.10
G&H	NiTi	6.09 ± 2.48 *	5.70 ± 1.81 *	1.03 ± 0.05	1.03 ± 0.05	2.74 ± 0.08 *	2.76 ± 0.07	0.51 ± 0.17 *	0.54 ± 0.18 *
	Steel	10.54 ± 7.82 *	9.24 ± 9.50 *	1.02 ± 0.06	1.01 ± 0.08	2.69 ± 0.11 *#	2.73 ± 0.14 #	0.36 ± 0.19 *	0.43 ± 0.20 *
Ormco	NiTi	5.79 ± 0.94 *	6.09 ± 1.03 *	0.97 ± 0.03 *	0.96 ± 0.03 *	2.71 ± 0.06 *#	2.69 ± 0.04 *#	0.48 ± 0.08 *	0.45 ± 0.08 *
	Steel	20.61 ± 12.08 *	23.99 ± 11.29 *	0.83 ± 0.09 *#	0.80 ± 0.07 *#	2.30 ± 0.21 *#	2.21 ± 0.20 *#	0.15 ± 0.10 *#	0.11 ± 0.09 *#

Statistically significant differences: *NiTi vs. steel of one manufacturer and #side 1 vs. side 2 of one feature.

## Data Availability

Data available from the authors: michal.sarul@gmail.com and kjurczysz@interia.pl.

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
