# Peer review of "Surface Evaluation of Orthodontic Wires Using Texture and Fractal Dimension Analysis"

_materials, 2021, doi:10.3390/ma14133688_

Round 1

Reviewer 1 Report

Reviewer Comments
1.   Please revise an abstract and follow the base form of an abstract journal.
2.   Check the rule on some words in your context.
3.   Graph is Fig.2. is too small, readers might be confused please clarify it.
4.   What does the table mean? The graph ‘AN and GN’ are not fixed on the page (the graph was plotted in excel you should choose another software). 
5.   What is the meaning from A-H on page 11? Some capital letters are bold and some are not.
6.   For all pictures need to be in the center of the page, table 3, figure 4 unacceptable.
To sum up, please follow the form of journal and revise all of it, too many errors

Author Response

Dear Reviewer,

Thank You for Your comments. We applied following changes:

  1.   Please revise an abstract and follow the base form of an abstract journal.

We changed our Abstract and tried to follow the rules of the Journal.

  1.   Check the rule on some words in your context.

We read our paper once again. Of course we found some mistakes, and we tried to correct them.

  1.   Graph is Fig.2. is too small, readers might be confused please clarify it.

We enlarged figure 2 to be more clarify.

  1.   What does the table mean? The graph ‘AN and GN’ are not fixed on the page (the graph was plotted in excel you should choose another software). 

Table 1 Legend was added: AN - Adenta NiTi, FN – Forestadent NiTi, ON - Ormco NiTi, GN - G&H NiTi, AS – Adenta Steel, FS - Forestadent Steel, OS – Ormco Steel, GS - G&H Steel.

Figure 3 is placed on the one page, legend is added and corrected.

  1.   What is the meaning from A-H on page 11? Some capital letters are bold and some are not.

It was corrected.

  1.   For all pictures need to be in the center of the page, table 3, figure 4 unacceptable.

All tables and figures were centered.

Best regards,

Authors.

Reviewer 2 Report

In the M&M section both of the wire have same names,  e.g. FN -FN, instead of FN -FS.

Figure 3 is partly visible.

There are too many references to make your point, and you used examples after corrosion, which is not the subject of your research.

Here are some additional comments for authors:
- in the Discussion section the sentences
from line 321-323 - there are standards for manufacturing of orthodontic wires, so, when you state "lack of standardization in wire surface topography", what do you mean?
-from line 326-329 - is confirmation of known facts. All research cited was performed on commercial wires.
-from line 332-336 - also, tests that you mentioned were already performed on commercial wires, therefore, the variability in surface is already taken into account.
-from line 377-379 did you mean specifically longitudinal scratches found in other studies, or surface roughness in general are associated with
increased corrosion and friction?

Author Response

Dear Reviewer,

Thank You for Your comments. We applied following changes:

In the M&M section both of the wire have same names,  e.g. FN -FN, instead of FN -FS.

We corrected following abbreviations:

  1. NiTi wires NiTi wires
  • Adenta – AN
  • Forestadent – FN
  • Ormco – ON
  • G&H – GN
  1. Chrome-nickel stainless steel wires:
  • Adenta – AS
  • Forestadent – FS
  • Ormco – OS
  • G&H – GS

Figure 3 is partly visible.

It was corrected.

There are too many references to make your point, and you used examples after corrosion, which is not the subject of your research.

We have decided to shorten this paragraph. We have only left the statements that relate to the overall effect of surface roughness on corrosion and ion release. We have left this part of the paragraph as a reference to the importance of the assessment of wire surfaces in our study in terms of their clinical properties.

We also shorten the References list.

Here are some additional comments for authors:
- in the Discussion section the sentences
from line 321-323 - there are standards for manufacturing of orthodontic wires, so, when you state "lack of standardization in wire surface topography", what do you mean?

We have changed the statement. Now it sounds: “Thus, the lack of homogeneity in the surface topography of the wires may affect the actual biomechanics of tooth displacement during fixed appliance treatment [34–39].” We hope, it is now more accurate.

-from line 326-329 - is confirmation of known facts. All research cited was performed on commercial wires.

We changed the statement for: “Thus, the presented studies describe one of the phenomena which make it impossible to assume that the friction values generated in the orthodontic wire / orthodontic bracket system and determined in laboratory tests are standard values for the given type of wire.” We believe taht nowi t is more accurate and states that our research confirms facts found by other authors and that we refer to their research standing in line with their results.

-from line 332-336 - also, tests that you mentioned were already performed on commercial wires, therefore, the variability in surface is already taken into account.

We changed the paragraph for:

“The lack of homogeneity of the surface topography of the tested orthodontic wires, confirmed in the study, may also be related to the degree of adhesion of the bacterial plaque [40-42]. The authors showed a lack of homogeneity in the surface structure even between the individual sides of the same wire. Moreover, in the presented study, using the analysis of textures and fractal dimensions, it was found that the factor of the wire producer has a greater impact on the homogeneity of the surface structure than the material from which the wire is made. It should be considered whether the above assumptions should not be taken into account when studying the degree of adhesion of bacterial biofilm to the surface of orthodontic wires. In addition, it is further evidence to what extent the perfection of the orthodontic component manufacturing process can be clinically relevant.”

We believe that now it is more accurate and understandable as postulate that the results obtained by us should be taken into account during subsequent studies in the field of biofilm adhesion.

-from line 377-379 did you mean specifically longitudinal scratches found in other studies, or surface roughness in general are associated with increased corrosion and friction?

As You could read we decided to shorten this paragraph and not to consider insightfully the mechanism of corrosion.

Best regards,

Authors.

Reviewer 3 Report

This study aims to evaluate the repeatability of orthodontic wire surface properties using fractal analysis. Please provide the null hypothesis in the Introduction!

In the discussion section, please present in a more precise manner the correlation between authors findings and available literature data!

Author Response

Dear Reviewer

Thank You for Your comments

We have moved the null hypothesis, that was in the Material and Methods section and now it is in the Introduction.

We have also added the paragraph in the Discussion. There are no publications in the literature that use fractal dimension or texture analysis method for examination of the surface of orthodontic wires, therefore our results cannot be directly related to other studies. However, in the paragraph added at Your request, we wrote that our research, using texture and fractal analysis, showed results similar to the results of other studies carried out with other research methods. We also described that in our study we extended the analysis to include differences in the surface structure of each individual wire. In addition, we referred to the fact that we mathematically proved that the wires of one of the manufacturers exhibit unevenness with a very uniform pattern. This is a factor that has not been described in other studies, and we hope You will find this reference satisfactory.

Sincerely Yours

Authors

Round 2

Reviewer 1 Report

The quality of the paper is very much improved.

It is already for publication.

Thank you.

Author Response

Dear Reviewer

Thank You for Your hard work, that let us to improve our manuscript.

We have checked our paper one more time and found some more minor language mistakes.

Thank You

And Sincerely Yours

Authors